# Moral Distress and Emotional Exhaustion in Healthcare Professionals: A Systematic Review and Meta-Analysis

**DOI:** 10.3390/healthcare13040393

**Published:** 2025-02-12

**Authors:** Alejandro Orgambídez, Yolanda Borrego, F. Javier Alcalde, Auxiliadora Durán

**Affiliations:** 1Department of Social Psychology, Universidad de Málaga, 29071 Malaga, Spain; fjalcalde@uma.es (F.J.A.); aduran@uma.es (A.D.); 2Department of Social and Educational Psychology, Universidad de Huelva, 21007 Huelva, Spain

**Keywords:** moral distress, emotional exhaustion, healthcare professionals, systematic review, correlation coefficient, meta-analysis

## Abstract

Background/Objectives: Moral distress is commonly experienced by healthcare professionals (i.e., nurses) as ethical conflict. Previous literature suggests that moral distress contributes to emotional exhaustion in these professionals. We aimed to synthesize and analyze studies that examined the relationship between moral distress and emotional exhaustion in healthcare professionals. Methods: Web of Science, Scopus, PubMed, Medline, and PsycInfo were used to search targeted studies written in English, Spanish, French, Italian, and Portuguese. The correlation coefficients of each study were extracted and converted into Fisher’s *z*. Finally, pooled *r* was calculated by Fisher’s *z* and standard error. The meta-analysis was performed with the R statistical program. Results: A total of 14 studies with 2425 healthcare professionals were included. The Moral Distress Scale Revised (MDS-R) and the Maslach Burnout Inventory (MBI) were the most used scales to measure moral distress and emotional exhaustion, respectively. The pooled correlation coefficient between moral distress and emotional exhaustion was 0.33 (*p* < 0.001, 95%CI: 0.27, 0.39). Conclusions: Moral distress is strongly correlated to emotional exhaustion in healthcare professionals. Future studies should focus on exploring the causal relationships between both variables as well as investigating potential moderators.

## 1. Introduction

The experience of moral distress is becoming particularly relevant in the work context of healthcare professionals, such as nurses, physicians, occupational therapists, or hospital social workers. Every day these professionals face situations where ethical issues often generate conflicts. That is, circumstances and decision making at work that potentially generate high levels of moral stress (MD) [1,2,3]. Although moral distress has been studied specifically during the COVID pandemic, this aspect is still present in the health context. In a recent study of Papatheodorou et al. [4] highlighted that up to 73.8% of the participants experienced moral distress in relation to their ability to provide care.

While there is consensus on when MD occurs, the definition of this phenomenon continues to be debated [5,6]. According to Morley et al. [5], the term MD has become an “umbrella term” that lacks conceptual clarity, referring unhelpfully to a wide range of phenomena and causes. It was originally described by Wilkinson [3] as “the psychological disequilibrium and negative feeling state experienced when a person makes a moral decision but does not follow through by performing the moral behavior indicated by that decision” (p. 16). Later, Jameton [2] distinguished three elements of ethical and moral conflicts—moral uncertainty, moral dilemma, and moral distress—defining the latter concept as “when one knows the right thing to do, but institutional constraints make it nearly impossible to pursue the right course of action” (p. 554). According to Morley et al. [5], at least three conditions must be met to refer to MD: (1) the experience of a moral event; (2) the experience of ‘psychological distress’; and (3) a direct causal relation between (1) and (2). Furthermore, professionals in various work contexts (e.g., an ICU nurse or a hospital social worker) may experience a wide range of moral distress experiences. However, what would be common to all of them would be the psychological stress response to ethical conflicts at work [5,6].

Healthcare professionals with high levels of moral distress often experience affective states of frustration, guilt, and anger [6,7,8]. Moral distress is associated with feelings of anger, disappointment, and guilt, resulting from the frustration of moral choice. Among the psychological consequences, depression, helplessness, psychological exhaustion, and sadness, among others, have been observed. Physical effects include insomnia, gastrointestinal problems, migraines, or excessive fatigue [5,8,9]. In that sense, various studies suggest that moral distress has consequences beyond the negative effects on the health and well-being of these professionals, also impacting quality of patient care and the functioning of the healthcare organization [5,10,11,12,13]. For example, participants in the qualitative study by Lewis et al. [14] expressed that the conflict between the interests and demands of patients, health care institutions, and family members and their values and beliefs as professionals negatively affected care decisions. As a result, these professionals felt exhausted and often fearful of being held personally accountable for possible adverse patient outcomes. Similar results were found in the study by Safari et al. [15], in which factors such as ethical considerations were negatively related to the quality of clinical care in an Iranian sample of nurses.

One of the most damaging effects of moral stress is getting burned out [9,12,13,16]. Burnout is defined as a psychological syndrome resulting from a prolonged response to stressors in the work context [17]. This syndrome is characterized by three symptoms or dimensions: emotional exhaustion, cynicism or depersonalization, and lack of professional efficacy [17]. Of the three elements of this syndrome, the emotional exhaustion dimension represents the core individual stress component of burnout. When health professionals feel emotionally exhausted, they report feeling overwhelmed at work and “empty” inside. In addition, these professionals feel weakened and drained, with insufficient energy to face another workday [10,16,18].

According to Maslach [17], value conflict, such as moral distress, can lead to situations that trigger burnout. The ethical conflict between what the professional wants to do and what he/she actually does, creates a psychological imbalance that, if sustained over time, can lead to burnout situations. When health professionals chronically experience this type of discrepancy in their ethical values, emotional exhaustion is likely to be the first visible symptom [7,9,16]. Furthermore, following the Job Demands-Resources model (JDR) [19,20], MD could be considered as a mismatch between job demands and the job and the personal resources required to cope with them. When the demand exceeds the resources (i.e., providing quality palliative care but not having the necessary organizational resources) and the situation cannot be handled ethically, it would trigger a process of health deterioration (i.e., lack of energy) that, if chronic, would eventually lead to burnout in these professionals [19,20].

Several publications have appeared in recent years documenting the relationship between MD and emotional exhaustion, including samples of different healthcare professionals (i.e., nurses, physicians, paramedics, etc.) [9,12,13,16]. However, a synthesis of the empirical evidence about the relationship between these two variables in the health context still needs to be completed. Given the influence of these variables on well-being, health, and quality of care in healthcare organizations, this review aimed to synthesize the results of studies focusing on the relationship between moral distress and emotional exhaustion in healthcare workers. Specifically, we proposed the following hypothesis:

**H1:** 
*Moral distress will be positively and significantly correlated to healthcare professionals’ emotional exhaustion.*


We consider that this review and meta-analysis will draw the attention of healthcare organizations managers to the relationship between MD and burnout. It will serve as a basis for future studies and interventions focused on the well-being and health of healthcare professionals, as well as the creation of ethical work environments where employees feel motivated and find meaning and purpose in their tasks.

## 2. Methods

### 2.1. Design

A systematic review was conducted according to the guidelines of the Joanna Briggs Institute [21], using the recommendations of the Preferred Items for Systematic Review and Meta-Analysis (PRISMA) guide as a reference for the selection and identification of studies [22,23]. The purpose of this systematic review was to synthesize the results of previous studies to gain a better understanding of the relationship between moral distress and the experience of emotional exhaustion in healthcare professionals. In addition, a meta-analysis was carried out with the numerical data of the included studies, which made it possible to obtain statistical coefficients that summarize the evidence previously analyzed, allowing a better interpretation of the relationship between the variables analyzed [24].

### 2.2. Search Strategy and Inclusion/Exclusion Criteria

The study was registered in the International Prospective Register of Systematic Reviews (PROSPERO) with the reference number CRD4202424568562 before the identification and selection of studies. The scientific databases Web of Science, Scopus, Pubmed, Medline, and PsycInfo were consulted, obtaining a total of 167 records during the search carried out between March and June 2024. The search terms “moral distress”, “MD”, “emotional exhaustion”, and “health care”, combined with the Boolean operators AND and OR, were included in each database with the following formula: “moral distress” OR “MD” AND “emotional exhaustion” AND “health care”. The bibliographic references of the selected studies were also consulted to obtain new studies that met the established inclusion/exclusion criteria. In those databases that include search filters related to language and publication type (i.e., Web of Science), the inclusion criterion [1] considered in this review was applied.

The inclusion criteria for the selection of studies were as follows: (1) articles published in peer-reviewed scientific journals in English, Spanish, French, and Portuguese; (2) studies that included samples of health professionals according to WHO (2019) (i.e., physicians, nurses, midwives, dentist, pharmacists, paramedical personnel, dietitians and nutritionists, physiotherapist, and various other therapy-related professions); (3) studies that collected a global measure of moral distress and emotional exhaustion, and reported coefficients of the relationship between the two variables (i.e., correlation coefficient, beta coefficient). The following exclusion criteria were established: studies with samples of health students, studies with qualitative and/or observational methodology, letters to the editor and journal editorials, and any review and/or meta-analysis.

### 2.3. Study Selection

The searches performed in the five databases resulted in a total of 167 records. The records were imported into the Rayyan web application [25] to automatically detect and remove duplicate studies, leading to the removal of 112 studies. Using the same web application, the titles and abstracts of the remaining 55 records were independently reviewed by two of the researchers, resulting in a total of 36 excluded studies that did not meet the inclusion criteria. In case of conflict, a third researcher was consulted to make the final inclusion/exclusion decision. After obtaining the 19 full-text papers, the studies were again reviewed independently by 2 members, excluding 6 records and leaving a total of 13 studies. Figure 1 presents the process followed according to the PRISMA guidelines [22,23].

### 2.4. Study Quality and Risk of Bias

Study quality was measured using the Quality of Survey Studies in Psychology (Q-SSP) tool [26]. The Q-SPP was developed by Protegrou and Hagger [26] to assess the quality of survey designs in the context of psychology. Since the variables moral distress and emotional exhaustion are assessed by self-report questionnaires, it was decided to use this tool to analyze the selected studies. The Q-SPP is composed of 4 dimensions characterized by 20 items: introduction (rationale and variables; 4 items), participants (sampling; 3 items), data (collection, analysis, measures, discussion; 10 items); and ethics (3 items). The tool allows us to obtain a percentage that reflects the quality of the study so that a value equal to or higher than 70% indicates an acceptable quality of the study [26].

After quality assessment by the three researchers (Appendix A), all the studies reached a minimum percentage of 70%, with a range between 73% and 84.21%. Finally, the following data were extracted from each record: authors, year, country, design, sample, aim(s), data collection, measurement scales, analysis, and main results.

### 2.5. Data Analysis

The statistical analyses were carried out with the R statistical program [27], using the meta and dmetar packages [24] for the transformation of Fisher’s *z* correlations. The Pearson and Spearman coefficients reported in each study were used to calculate the effect size. In addition to a value of *r* retrieved from each paper, the conversion between *r* and other statistics (i.e., beta regression coefficients) was computed in those studies in which only beta coefficients were reported. We followed Peterson and Brown’s orientations [28] for the conversion of beta coefficients into correlation coefficients according to the following formula: *r* = *β* + 0.05(*λ*). The lambda coefficient (*λ*) assumes a value of 1 when beta is positive and a value of 0 (zero) when it is negative.

We used a random-effects model because we assumed that some degree of between-study heterogeneity is expected, given the diversity of the samples and methodological aspects of each study [24]. To assess heterogeneity, we utilized the following measures: Cohran’s *Q* and *tau*^2^, *I*^2^ statistics with confidence intervals, and prediction intervals (PIs). *I*^2^ values higher than 50–70% are indicative of moderate-high heterogeneity, being recommended in these cases to check for outliers and influential cases. The effect (g) can be considered robust if the value 0 (zero) is not included in the calculated prediction interval [24].

We produced a funnel plot to measure publication bias across studies, looking at the relationship between precision and observed effect size of studies (Figure 2). A funnel plot is a graphic technique for assessing potential publication bias where “an unbiased sample would ideally show a cloud of data points that is symmetric around the population correlation coefficient and has the shape of a funnel” [29] (p. 686). In addition, Egger’s regression test was calculated to check the asymmetry of the funnel plot, obtaining a coefficient of 1.695 (9%CI: −1.78, 5.17), *t* = 0.96, *p* = 0.36. In this sense, Eggers’ test does not indicate the presence of funnel plot asymmetry.

## 3. Results

### 3.1. Study Characteristics

A total of 14 studies, published between 2004 and 2024, were included in the present review. Appendix A shows the characteristics of the selected studies. All studies applied an analytical approach: almost all were cross-sectional, and only one included a longitudinal design (six time-points cohort design) [30]. The studies by Grasso et al. [31] and Ohnishi et al. [32] focused specifically on measurement scales’ validation, establishing the relationship with emotional exhaustion as an indicator of criterion validity.

The studies were performed on samples from different countries. Six studies were conducted in Europe: four in Italy [9,31,33,34], one in the Netherlands [12], and one in Cyprus [35]. Six investigations were conducted in the Americas: three in the USA [36,37,38], two in Canada [30,39], and one in Brazil [11]. Finally, the study by Nassehi et al. [13] was conducted in Iran and the study by Ohnishi et al. [32] in Japan. The total number of participants in the selected studies was 2.425. Most of the participants were nurses (1500; 61.85%). Regarding the type of analysis in the 14 studies, the statistical analyses used to examine the relationship between moral distress and emotional exhaustion were diverse: correlations (i.e., Pearson correlation analysis), multiple linear regression models, logistics models, mediation models, and moderation models.

The most common measure applied to assess moral distress was the Moral Distress Scale Revised [40], which was present in eight studies: Carletto et al. [9], Christodoulou-Fella et al. [35], Fumis et al. [11], Grasso et al. [31], Kellish et al. [36], Kok et al. [12], Maffoni et al. [34], and Sajjadi et al. [39]. The remaining six studies used the following scales: Moral Distress Scale by Corley et al. [1] (two studies: Meltzer and Huckabay [37]; Rushton et al. [38]); Moral Distress Scale for Psychiatric Nurses by Ohnishi et al. [32] (one study: Delfrate et al. [33]); Measure of Moral Distress for Healthcare Professionals (MMD-HP) by Epstein et al. [41] (one study: Maunder et al. [30]); and Moral Distress Scale by Jafari et al. [42] (one study: Nassehi et al. [13]).

All but one study used the emotional dimension of the Maslach Burnout Inventory (MBI) [43]. In Christodoulou-Fella et al.’s [35] study, exhaustion was assessed with a single question using a 10-point Likert scale.

### 3.2. Meta-Analysis of the Correlation Between Moral Distress and Moral Exhaustion

The relationship between moral distress and emotional exhaustion in the selected studies ranged from 0.15 to 0.54. No significant association between both variables was observed in the study of Grasso et al. [31]: *r* = 0.15, *p* > 0.05. The meta-analysis showed that moral distress and emotional exhaustion were positively and significantly related (*g* = 0.33, *p* < 0.001; 95%CI: 0.27, 0.39) (Figure 3). The total number of participants included in the meta-analysis was 2.245 (*k* = 14). The *Q-value* was 29.20 (*p* < 0.01). The between-study heterogeneity variance was estimated at *tau*^2^ = 0.0081 (95%CI: 0.0337, 0.1835), with an *I*^2^ value of 55.5% (95%CI: 18.9%, 75.6%). The prediction interval ranged from *g* = 0.13 to 0.50, indicating that a positive correlation may be expected for future studies. After running the diagnosis of influential cases and the graphic display of heterogeneity [44,45], no study was detected as an outlier and/or influential case.

## 4. Discussion

The present study identified a total of 14 studies that examined the relationship between moral distress and emotional exhaustion in healthcare professionals. All studies were of acceptable quality according to the criteria set by the Q-SPP tool [26]. Most of the participants in the studies were nurses and the research was conducted in different countries in America, Asia, and Europe, especially in Italy. The most used scale to measure moral distress was the Moral Distress Scale Revised [40], applied in eight of the fourteen studies, while the MBI [43] was the most used questionnaire to assess emotional exhaustion. Variability in the sample composition (i.e., nurses, physicians, social workers, etc.), countries, and healthcare systems, as well as cultural differences in professional ethics, may explain some of the heterogeneity present in the review.

The meta-analysis conducted showed a significant and moderate relationship between moral distress and emotional exhaustion among health professionals, supporting the hypothesis presented in this study. These findings are broadly consistent with the theoretical assumptions of the Job Demands-Resources model [19,20], which states that moral distress is the result of an imbalance between job demands and the personal and job resources to cope with them. The inability to provide ethically adequate health care due to a lack of resources (e.g., lack of ICU beds, few respirators in the unit, etc.) would, in the long run, lead to a spiral of deterioration in the physical and psychological health of the professional, manifesting as high levels of emotional exhaustion [19,20]. In that sense, Maslach [17] declares that ethical conflicts are one of the main antecedents of burnout. If health professionals chronically suffer from this type of moral stressors or ethical conflicts at work, emotional exhaustion will likely emerge as a basic individual response to chronic stress.

Given the impact of moral stress on health, psychological well-being, and quality of care, the results of this study highlight the importance of the development and implementation of interventions in social and health organizations by the managers and heads of these institutions, especially for those professionals who are at higher risk of being in morally stressful situations (i.e., professionals in intensive care units, oncology, etc.) [12,35]. As Gustavsson et al. [46] stated, the goal should not be to eliminate employee moral stress, a normal reaction to moral issues in healthcare settings, and is impossible to eradicate given the complex nature of the profession. A more accurate aim would lead us to develop a culture of moral resilience, which would help us to manage and mitigate moral distress [47]. Moreover, it would become an essential protective factor against high levels of quiet quitting, job burnout, and turnover intention [48].

Different categorizations can be used as a framework to guide the intervention proposals. While Salas-Bergués et al. [49] identified personal, organizational, job relation, situational, and end-of-life care factors in developing burnout and moral distress, Hancock et al. [50] categorized this complexity in organizational issues, exposure to high-intensity situations, and poor team experiences.

So, alongside interventions aimed at explaining the individual’s resources to cope with the situations in daily work, such as stress reduction strategies based on mindfulness, self-enhancement awareness, or strong communication [49], there is a clear need to design interventions regarding organizational factors linked to lack of ethical and supportive leadership and organizational structures (i.e., sufficient staff and resources) to reduce the likelihood of morally distressing situations. In fact, Papatheodorou et al. [4] stated that high percentages of healthcare professionals believe that important factors that would alleviate MD are more staff resources (81.6%), less bureaucracy (45.2%), greater emotional and psychological support (44.6%), and more training (39.5%).

Those professionals with symptoms of moral stress and emotional exhaustion need to be prioritized, supported, and clinically supervised to avoid further deterioration of their physical and psychological health. Psychological counseling services in the workplace and social support groups (i.e., among colleagues) are also tools that allow the professional to deal more adequately with the possible ethical and moral conflicts to which he/she may be exposed. In this sense, tangible, emotional, and ethically coherent organizational and social support provides an adequate response to the symptoms of moral stress.

Similarly, greater awareness of the experience of moral distress and its consequences within organizations, the development of ethical organizational cultures, and training in coping and leadership skills are also strategies that can be implemented to prevent and reduce this type of distress. In this sense, discussion of ethics in clinical settings can be positive and precedes awareness of how to meet the needs of patients/users. Social and healthcare organizations can consider providing ethical consultation and discussion, encouraging support for using ethical resources, and promoting appropriate supervision [49]. As Severinsson [50] indicated, it is essential to recognize healthcare professionals’ needs, including both emotional support and the right to systematic clinical supervision to help them reflect on their work and interpret the needs of patients/users. Adopting perspective-building and adaptation strategies, including active, reflective, and structural supports, may also help professionals to benefit from improved and accessible formal support. In this regard, strategies focused on improving the work environment and teamwork would play an essential role. Supportive interventions to improve professionals’ empowerment, autonomy, and ethical knowledge could help to mitigate moral distress and burnout. Ethical leaders, as mentors and role models, access to ethical resources, continuing ethics education, and specific support mechanism, such as ethics committees, could help healthcare professionals grow after experiencing morally charged events [49].

### Limitations and Future Research

Several limitations need to be acknowledged. First, almost all the studies are cross-sectional in nature, which may affect the generalization of the findings. In addition, the review focuses on the relationship between moral distress and emotional exhaustion, which makes it impossible to establish cause-and-effect relationships given the correlational nature of the selected studies. Given that most of the studies reviewed were correlational and cross-sectional in nature, it is recommended that future research prioritizes the development of longitudinal research designs, comprising multiple data collection points over time. This approach would facilitate a more comprehensive investigation of the underlying mechanisms linking these occupational phenomena by analyzing possible moderators/mediators present in the relationship between the two variables. Second, the inclusion criteria related to the language of publication may have introduced a bias in the selected studies, as they may have excluded research published in Asian and African countries. Third, only reported results of the studies and not the original data were analyzed. Finally, qualitative studies that could complement the findings of this study with the lived experiences of these professionals were not reviewed. Nevertheless, future qualitative reviews on moral distress and emotional exhaustion (i.e., meta-ethnographies) would help in interpreting the personal and work processes experienced in situations of moral conflict.

Nevertheless, relevant strengths can also be identified, as a rigorous search for and selection of studies, an in-depth analysis, results based on quality research, and implications for professional practice.

## 5. Conclusions

This review provides evidence for a positive relationship between moral distress and emotional exhaustion in health professionals. Further longitudinal and experimental studies would be needed to better understand the processes and mechanisms involved in both occupational phenomena. These findings highlight the need for workplace interventions and strategies to reduce moral distress and, as a consequence, emotional exhaustion in healthcare organizations.

## Figures and Tables

**Figure 1 healthcare-13-00393-f001:**
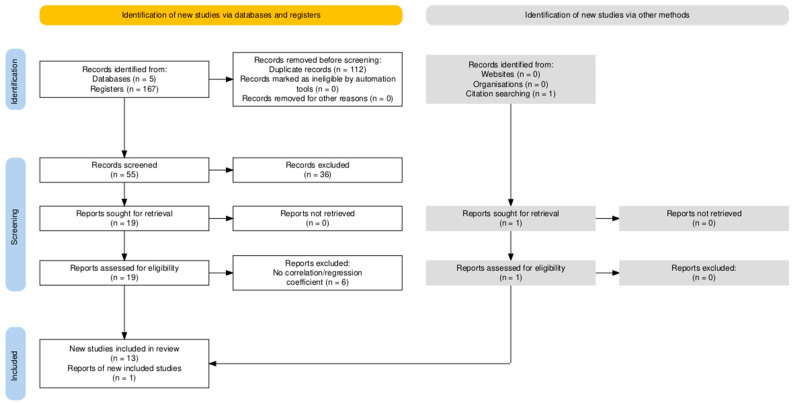
Flow diagram of literature search.

**Figure 2 healthcare-13-00393-f002:**
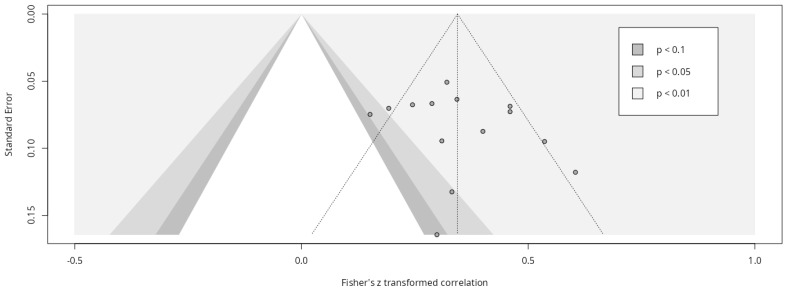
Funnel plot.

**Figure 3 healthcare-13-00393-f003:**
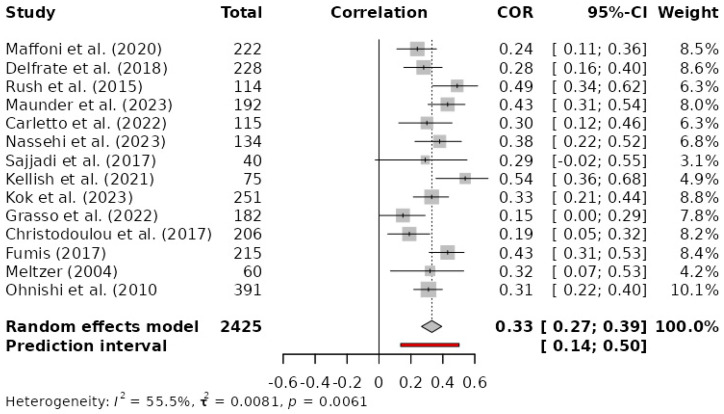
Random effects forest plot for moral distress and emotional exhaustion relationship [9,11,12,13,30,31,32,33,34,35,36,37,38,39].

## Data Availability

Data are contained within the article or Appendix A.

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
