# Peer review of "Moral Distress and Emotional Exhaustion in Healthcare Professionals: A Systematic Review and Meta-Analysis"

_healthcare, 2025, doi:10.3390/healthcare13040393_

Round 1

Reviewer 1 Report

Comments and Suggestions for Authors

Comments

Introduction:

Expand the introduction with examples of how moral distress impacts the quality of patient care, citing recent studies to support these observations.

Methodology: Provide more details on the rationale for excluding qualitative studies and discuss potential limitations resulting from this decision.

Results: Include tables summarizing the characteristics of the studies, such as sample sizes and participant contexts, for clearer representation of the findings.

Discussion: Expand the discussion on cultural and contextual implications. Addressing regional differences between the included studies will add greater robustness to the analysis.
Conclusion:
Include recommendations for future research, such as conducting longitudinal studies and analyzing potential moderators.

Other Considerations:

 Perform subgroup analyses to identify significant contextual variations. For example:

  • Compare healthcare professionals working in intensive care units with those in outpatient clinics;
  • Assess whether the relationship between moral distress and emotional exhaustion differs between regions with well-structured versus less-structured healthcare systems.

 The study does not examine potential moderators that could influence the relationship between moral distress and emotional exhaustion, such as workload, organizational support, or professional experience. Consider using meta-regression analyses to explore the impact of moderator variables on the results.

The article does not mention conducting formal tests for publication bias, such as Egger’s test or creating funnel plots. Implementing these tests could help identify and report potential publication biases, which are common in systematic reviews and meta-analyses.

The study lacks a detailed analysis of the demographic or contextual characteristics of the participants. Summarize demographic and contextual data (e.g., average age, gender, country, type of healthcare setting) in tables or graphs to provide a clearer picture of the study sample.

Author Response

# Introduction:
Comments 1: Expand the introduction with examples of how moral distress impacts the quality of patient care, citing recent studies to support these observations.
Response 1: Thanks for the suggestion. We have added recent research (i.e., Lewis et al., 2025; Safari et al., 2024) describing the impact of moral stress on quality of care. Page 3, first paragraph.

# Methodology:
Comments 2: Provide more details on the rationale for excluding qualitative studies and discuss potential limitations resulting from this decision.
Response 2: Thank you for your comments. The reason for excluding qualitative studies is determined by the conduct of the meta-analysis, which requires studies that report a coefficient of association (i.e., Pearson's r) between the two variables. Only quantitative studies report this type of statistical coefficient, which establishes qualitative studies as one of the exclusion criteria. In section 4.1. Limitations and Future Research, we pointed out potential limitations resulting from this decision.

#Results:
Comments 3: Include tables summarizing the characteristics of the studies, such as sample sizes and participant contexts, for clearer representation of the findings.
Response: Table S2 (Supplementary Data) shows the characteristics of the selected studies: authors, year, country, design, participants, aim(s), data collection and scales, and main results. The Table S2 is cited on the section 3.1. Study characteristics.

# Discussion:
Comments 4: Expand the discussion on cultural and contextual implications. Addressing regional differences between the included studies will add greater robustness to the analysis.
Response 4: Thank you for your comments. Since we did not perform subgroup meta-analyses by region, we are unable to discuss findings that are not included in the manuscript. Regarding the contextual implications, special emphasis was placed on providing practical strategies for the prevention of moral distress from different perspectives, including both organizational and clinical contexts.

#Conclusion:
Comments 5: Include recommendations for future research, such as conducting longitudinal studies and analyzing potential moderators.
Response 5: We referred to the need for future longitudinal studies in Section 5, Conclusions. We have added the study of possible moderators/mediators as one of the recommendations for future studies.

#Other Considerations:

Comments 6: Perform subgroup analyses to identify significant contextual variations. For example:
    Compare healthcare professionals working in intensive care units with those in outpatient clinics;
    Assess whether the relationship between moral distress and emotional exhaustion differs between regions with well-structured versus less-structured healthcare systems.
Response 6: Thank you very much for the suggestion. We consider of particular importance for future publications to explore possible differences in the moral stress-emotional exhaustion relationship in different health systems.

Comments 7: The study does not examine potential moderators that could influence the relationship between moral distress and emotional exhaustion, such as workload, organizational support, or professional experience. Consider using meta-regression analyses to explore the impact of moderator variables on the results.
Response 7: Thank you for your comment. The main objective of the review was to test the relationship between moral stress and emotional exhaustion. We agree that future reviews should focus on analyzing possible moderators/mediators of the relationship between the two variables.

Comments 8: The article does not mention conducting formal tests for publication bias, such as Egger’s test or creating funnel plots. Implementing these tests could help identify and report potential publication biases, which are common in systematic reviews and meta-analyses.
Response 8: Eggers' test and funnel plots (Figure 2) are included in the paper, in the section 2.5. Data analysis.

Comments 9: The study lacks a detailed analysis of the demographic or contextual characteristics of the participants. Summarize demographic and contextual data (e.g., average age, gender, country, type of healthcare setting) in tables or graphs to provide a clearer picture of the study sample.
Response 9: Information of the demographic characteristics, samples, design, countries, etc. are shown in Table S2 (Supplementary Data) - Characteristics of the included studies.

Reviewer 2 Report

Comments and Suggestions for Authors

The manuscript is well-crafted, well-structured throughout; the text is understandable and the references are appropriate. It is not clear to me why the manuscript has been identified as a review when it is a full-fledged research paper (in fact, the structure of the manuscript is of a typical research paper, with intro, materials and methods, results, discussion and conclusions). For this reason I suggest fewer revisions, although for me it may already be ready for publication. Abstract: The study context is moral distress, commonly experienced by health care professionals (e.g., nurses) as an ethical conflict. Previous literature suggests that moral distress contributes to the emotional exhaustion of these professionals. The aim of the paper was to synthesize and analyze studies that have examined the relationship between moral distress and emotional exhaustion in the health professions. Perhaps that is why it is considered a review, but the structure of the manuscript is typical of a research paper. The authors need to clarify this passage. The search method as well as the materials are appropriate because Web of Science, Scopus, PubMed, Medline and PsycInfo were used to search for focused studies written in English, Spanish, French, Italian and Portuguese. Statistical analyses are functional and complete for the research objectives. Fourteen studies with 2425 health care workers were included, and the scales investigated are sufficient, reaching the conclusion that moral distress is moderately related to emotional exhaustion in health care workers. Results, discussions and conclusions are consistent and complete. Overall, it is an excellent work that deserves publication.

Author Response

Comment 1: The manuscript is well-crafted, well-structured throughout; the text is understandable and the references are appropriate. It is not clear to me why the manuscript has been identified as a review when it is a full-fledged research paper (in fact, the structure of the manuscript is of a typical research paper, with intro, materials and methods, results, discussion and conclusions). For this reason I suggest fewer revisions, although for me it may already be ready for publication. Abstract: The study context is moral distress, commonly experienced by health care professionals (e.g., nurses) as an ethical conflict. Previous literature suggests that moral distress contributes to the emotional exhaustion of these professionals. The aim of the paper was to synthesize and analyze studies that have examined the relationship between moral distress and emotional exhaustion in the health professions. Perhaps that is why it is considered a review, but the structure of the manuscript is typical of a research paper. The authors need to clarify this passage. The search method as well as the materials are appropriate because Web of Science, Scopus, PubMed, Medline and PsycInfo were used to search for focused studies written in English, Spanish, French, Italian and Portuguese. Statistical analyses are functional and complete for the research objectives. Fourteen studies with 2425 health care workers were included, and the scales investigated are sufficient, reaching the conclusion that moral distress is moderately related to emotional exhaustion in health care workers. Results, discussions and conclusions are consistent and complete. Overall, it is an excellent work that deserves publication.

Response 1: Thank you for your comment and positive feedback. We have followed the structure proposed by the journal for a systematic review and meta-analysis.

Reviewer 3 Report

Comments and Suggestions for Authors

Overall - very well written!

Suggested edits:

Abstract:  Lines 21-23 - Reword:  Future studies should focus on exploring the ......as well as investigating potential moderators.

Introduction: 

Lines 32-34 - briefly describe that the study was about. - In a recent study of xxxxx, .... 

Line 45 - choose another word for "speak"

Line 51 -  you have already noted that MD will be used for moral distress so correct throughout the paper.  Except for Line 52 - spell out Moral Distress as it is beginning a sentence.

Line 58-59 .....quality of patient care and the functioning of the healthcare organization.

Line 64 - need citation after the sentence that ends with "efficacy"

Line 66 - need citation for sentence that ends with "inside"

Line 67 - reword sentence - In addition, these professionals.......

Line 69 is moral stress and moral distress the same?  Consider using the same term throughout - or define the difference.  

Line 83 - check font size of citation

Line 89-90 - Not sure that a hypothesis is customary for a systematic review - if it remains - consider rewording to positive correlation.

Line 91-93.  ... will draw attention of healthcare organizations' manages to the relationship of MD to healthcare provider burnout.

Line 104 - consider .... numerical data of the included studies....

Line 137 - not clear what 6 "registers" is 

Author Response

Comments 1: Overall - very well written!

Suggested edits:

Abstract:  Lines 21-23 - Reword:  Future studies should focus on exploring the ......as well as investigating potential moderators.

Introduction: 

Lines 32-34 - briefly describe that the study was about. - In a recent study of xxxxx, .... 

Line 45 - choose another word for "speak"

Line 51 -  you have already noted that MD will be used for moral distress so correct throughout the paper.  Except for Line 52 - spell out Moral Distress as it is beginning a sentence.

Line 58-59 .....quality of patient care and the functioning of the healthcare organization.

Line 64 - need citation after the sentence that ends with "efficacy"

Line 66 - need citation for sentence that ends with "inside"

Line 67 - reword sentence - In addition, these professionals.......

Line 69 is moral stress and moral distress the same?  Consider using the same term throughout - or define the difference.  

Line 83 - check font size of citation

Line 89-90 - Not sure that a hypothesis is customary for a systematic review - if it remains - consider rewording to positive correlation.

Line 91-93.  ... will draw attention of healthcare organizations' manages to the relationship of MD to healthcare provider burnout.

Line 104 - consider .... numerical data of the included studies....

Line 137 - not clear what 6 "registers" is 

Response 1: Thank you for your comments and positive feedback. We have included all your suggestions in the text.

Round 2

Reviewer 2 Report

Comments and Suggestions for Authors

The changes made make the manuscript ready for publication.